# The Use of Shells of Marine Molluscs in Spanish Ethnomedicine: A Historical Approach and Present and Future Perspectives

**DOI:** 10.3390/ph16101503

**Published:** 2023-10-23

**Authors:** José A. González, José Ramón Vallejo

**Affiliations:** 1Grupo de Investigación de Recursos Etnobiológicos del Duero-Douro (GRIRED), Facultad de Biología, Universidad de Salamanca, E-37071 Salamanca, Spain; 2Departamento de Anatomía Patológica, Biología Celular, Histología, Historia de la Ciencia, Medicina Legal y Forense y Toxicología, Área de Historia de la Ciencia, Facultad de Medicina, Universidad de Cádiz, E-11003 Cádiz, Spain; joseramon.vallejo@uca.es

**Keywords:** cuttlebone, nacre, seashells, Spanish ethnomedicine, historical data, pharmacology, health humanities

## Abstract

Since ancient times, the shells of marine molluscs have been used as a therapeutic and/or prophylactic resource. In Spain, they were part of practical guides for doctors or pharmacists until the 19th century. In general, seashells were prepared by dissolving in vinegar and were part of plasters or powders used as toothpaste, or to treat dyspepsia, heartburn and leprosy. Thus, the nacre or mother-of-pearl of various molluscs was regularly used in the Royal Colleges of Surgery and in hospitals during the times of the Cortes of Cadiz, as a medicine in galenic preparations based on powders. In contemporary Spanish ethnomedicine, seashells, with a high symbolic value, have been used as an amulet to prevent cracks in the breasts and promote their development during lactation, to avoid teething pain in young children, to eliminate stains on the face or to cure erysipelas. But, as in other countries, products derived from seashells have also been empirically applied. The two resources used traditionally have been the cuttlebone, the internal shell of cuttlefish and the nacre obtained from the external shells of some species. Cuttlebone, dried and pulverised, has been applied externally to cure corneal leukoma and in dental hygiene. In the case of nacre, a distinction must be made between chemical and physical remedies. Certain seashells, macerated in lemon juice, were used in coastal areas to remove spots on the face during postpartum. However, the most common practice in Spain mainland was to dissolve mother-of-pearl buttons in lemon juice (or vinegar). The substance thus obtained has been used to treat different dermatological conditions of the face (chloasma, acne), as well as to eliminate freckles. For the extraction of foreign bodies in the eyes, a very widespread traditional remedy has been to introduce small mother-of-pearl buttons under the lid. These popular remedies and practices are compared with those collected in classic works of medicine throughout history, and data on the pharmacological activity and pharmaceutical applications of the products used are provided. The use of cuttlebone powders is supported by different works on anti-inflammatory, immune-modulatory and/or wound healing properties. Nacre powder has been used in traditional medicines to treat palpitations, convulsions or epilepsy. As sedation and a tranquilisation agent, nacre is an interesting source for further drug development. Likewise, nacre is a biomaterial for orthopaedic and other tissue bioengineering applications. This article is a historical, cultural and anthropological view that can open new epistemological paths in marine-derived product research.

## 1. Introduction

In contemporary Spanish ethnomedicine, only a single use-report concerning the medical use of an entire marine mollusc has been documented. So, in Bedarona (Biscay), when they had warts on their hands, they collected limpets (genus *Patella*, Gastropoda, Patellidae) on the rocks along the coast. With each of these snails they rubbed a particular wart several times, and then killed them by poking them with a stick, in the belief that when the limpets rotted the warts would disappear [1]. However, in Spain, the most important therapeutic resource for this group of invertebrate animals has been their shells.

### Symbolisms Associated with Shells

There is evidence and data on the use of seashells in different societies since time immemorial. Thus, we find studies on archaeozoology that show that the coastal malacofauna (molluscs) was used in the eastern part of the Iberian Peninsula from the Neolithic Age to the end of the Bronze Age for the manufacture of ornamental elements, and with a value linked to the world of symbolism and magic [2]. In some cases, they were personal attributes of an ornamental, aesthetic, ideological or symbolic nature of the amulet type [3,4]. It is evident that the world of beliefs, symbols and amulets form part of the first medical manifestations and that they have endured through the history of therapeutics [5,6]. In this way, since ancient times, the shells of marine molluscs have been used in the Mediterranean region as an apotropaic amulet to preserve both humans and domestic animals from harm and disease [5,7].

For this reason, it is very interesting to combine the methodologies of anthropology and history to study the traditions of popular culture. In the case of the symbolism associated with seashells, we can obtain a whole series of cultural interpretations of the historical experience. For example, because of its clear resemblance, shape and colouring, reminiscent of a woman’s vulva, the shell combines sexual symbolism with the idea of procreation and fertility. All this makes it the attribute of the goddess of love: Aphrodite-Venus materialises by emerging from the foam of the ocean and, placed in a shell, is carried to the mainland [8,9,10,11].

Paleo-Christian art made the empty shell an image of the departure of the soul to immortality [10], a symbol of birth into the future life, of the resurrection [8]. Christian symbolism saw the shell as the image of the sepulchre that embraces man after death before he can be resurrected [9]. As a symbol of fertility, the shell also became a symbol and attribute of the Virgin Mary [9,11].

## 2. Taxonomy and Uses in Folk Medicine

### 2.1. Magical Uses

Popularly known as the *caracol de viento* (“wind snail”), hung around the neck, it was used in various Spanish regions as a superstitious remedy to prevent teething pain in small children, to cure erysipelas and to remove spots from the face [12,13].

In 1883, in his work *Supersticiones populares recogidas en Andalucía y comparadas con las portuguesas* (“Popular superstitions collected in Andalusia and compared with those of Portugal”), the folklorist Alejandro Guichot y Sierra noted that he had heard this usage said in public to a shell seller in Seville, a seller who had a tarpaulin stretched on the ground on which piles of various seashells were displayed, and who was listened to by a large group of curious onlookers [14].

On the other hand, it is worth mentioning that there are many traditional methods used in Spain to prevent cracks in the breasts during breastfeeding, and different amulets have also been used which, according to popular belief, prevent them from appearing.

The *similia similibus curantur* (“likes are cured by likes”) principle can be applied to almost all of the amulets used for this purpose, since they are objects which, because of their resemblance to healthy and voluminous breasts, were used to prevent them from deteriorating and to facilitate their firming and resistance to the effects of breast-feeding [15,16]. In the present work we must point out the use of the shells of small marine molluscs, especially those of some representative species of the family Cypraeidae, commonly named cowries, and the shells of cockles (the common cockle, *Cerastoderma edule* (Linnaeus, 1758)), but also the use of those commonly known as *buguinas*, i.e., large and edible marine gastropods as the common whelk (*Buccinum undatum* Linnaeus, 1758; Buccinidae) or the knobbed triton (*Charonia lampas* (Linnaeus, 1758); Charoniidae) [17].

### 2.2. Empirical Remedies

In contemporary Spanish ethnomedicine, products derived from seashells have been used—applied in an empirical way—specifically the cuttlebone, the internal shell of cuttlefish (*Sepia officinalis* Linnaeus, 1758; Cephalopoda, Sepiidae) and the nacre (or mother-of-pearl) from the shells of certain species (e.g., *Pinna nobilis* Linnaeus, 1758; *Haliotis tuberculata* Linnaeus, 1758) (Figure 1).

#### 2.2.1. The Cuttlebone in Spanish Ethnomedicine

Known in Spain by the names *sepión*, *jibión* or *hueso de la jibia*, the cuttlebone has been a zootherapeutic resource in Spanish ethnoveterinary medicine. Dried and pulverised, the cuttlebone has been applied externally (topical use) to cure corneal ulcers, infectious bovine keratoconjunctivitis (“pinkeye”) and udder injuries, and by its internal use to treat aphthous fever in cattle [18].

In the field of ethnomedicine, we find a reference to its use in the treatment of corneal leucoma. So, in Tenerife (Canary Islands), lesions appearing on the cornea as a result of a blow or injury were treated by applying the powder obtained from grinding this internal shell to the affected eye [19,20].

In Asturias, this powder was used to clean teeth, for dental hygiene [21].

#### 2.2.2. Nacre in Spanish Ethnomedicine

The seashells richest in nacre were used in folk dermatology; they constituted a popular remedy for removing spots on the face that can be classified as “chemical”. Certain shells were macerated in lemon juice.

In Palamós (Gerona), women would fight against postpartum spots (melasma or chloasma) by applying a white paste obtained by macerating *curculles de nacre* (mother-of-pearl shells) in lemon juice for 9 or 10 days [22].

Nevertheless, due to the difficulty of obtaining mother-of-pearl shells far from the coast, or simply because it was a readily available resource, mother-of-pearl buttons were also dissolved in lemon juice or vinegar to obtain the remedy. So, in Extremadura, different types of pimples and acne were treated by applying the paste obtained from dissolving several mother-of-pearl buttons in lemon juice or vinegar to the affected areas [23,24,25].

To cure eczema, in the Canary Islands, they put a mother-of-pearl button in lemon juice and left the container in the open air all night. By dawn, the button had dissolved, and the substance obtained could then be used [26,27].

To make freckles, considered to be ugly or rather unattractive, disappear from the face, in the provinces of Cádiz [28], Badajoz [29] and Málaga [30], but also in far off Navarre [1], the face was washed with the mixture obtained by macerating one or more mother-of-pearl buttons in lemon juice. The glass was left in the open air overnight, and the following morning, when the buttons had melted, the liquid obtained was applied to the area where the freckles were. The treatment had to be repeated for at least a week.

The other field of ethnomedicine in which seashells very rich nacre were used is popular ophthalmology. In this case, we have to talk of a “physical-mechanical” type of remedy for the removal of foreign bodies from the eyes.

The nutlets (mericarps) of salvia, principally of the species *Salvia verbenaca* L., have been the most widespread remedy in Spain for the removal of foreign bodies from the eyes. Small in size (1.6–2.4 mm), producing mucilaginous and very soft substances, and popularly known as “eye-cleaners”, they have also been used to clean eyes in other parts of the Mediterranean region [31,32,33]. When a foreign body (an eyelash, dust, straw, etc.) enters the eye, a nutlet was placed under the lid to force it out. This practice was especially important in past decades when hand threshing and winnowing were still carried out in our country [34,35].

Continuing the importance of their use, we should also mention in this review the mechanical use of small mother-of-pearl buttons to remove foreign bodies from the eyes. So, if something got into the eye while working in the field, a small mother-of-pearl button was placed under the eyelid. This practice has been documented in Extremadura [24,36,37,38,39] and in the provinces of Alicante [40] and Soria [41].

In contrast, in Humanes (Guadalajara), they would insert a mother-of-pearl button into the eye when they had a stye, “to break its head”, i.e., with the aim of reducing the inflammation and releasing the pus that these painful eye conditions usually contain [42].

## 3. Medical History: The Seashells and the Cuttlebone between Popular Pharmacology and Science

Popular therapeutics and scientific medicine have shared pharmacological resources throughout history, entering into opposition from cultural, hegemonic or scientific positions according to their role and competencies in each era. However, medicines based on animal resources have undergone transfers from one medical system to another. In fact, we currently have the case of animals such as leeches or maggots, i.e., the larvae of *Lucilia sericata* (Meigen, 1826) fly species, which have once again become part of current hospital medicine [43,44]. On the other hand, classic studies on popular medicine carried out in the Basque Country (Spain) and other regions of Europe, allow us to affirm that 50% of these practices have their origin in scientific medicine, 35% have magical and religious components, 20% would be empirical remedies and 5% would derive from primitive medicines [45,46]. Moreover, depending on the epistemological context in which studies seeking to understand health and therapies are framed, approaches and concepts have been developed ranging from ethnomedicine, to medicalisation or other more recent concepts such as medical pluralism or traditional ecological knowledge related to medicine [47,48,49,50]. In this sense, there could still be debates to define and use these terms or ignore other perspectives. In our opinion, it is important to break the disciplinary gap as a consequence of the complexity of fields such as health and sustainability, avoiding disciplinary tensions and unconnected, fragmented research based on stereotypes [51,52]. Thus, it is important to consider transdisciplinary approaches that can support research on drug candidates and zootherapeutic products in line with biodiversity and the “One Health” philosophy across multiple disciplines. It is therefore relevant to consider the uses of seashells, mother-of-pearl and cuttlefish bone in different historical periods according to the science of the time and to analyse their current perspectives from a multidimensional perspective.

### 3.1. From Classical Civilisation to the Middle Ages

Undoubtedly, we must start from the Greco-Roman world as it helps us to understand the roots of its ethnomedical use; however, in the present work we are mainly interested in the use in medicine from the Modern Age. That is, when official medicine began to approach new medical mentalities and the new paradigms of the natural and experimental sciences.

As we have already described in a previous work [18], in his encyclopaedia and pharmacopoeia *De Materia Medica*, Pedanius Dioscorides (ca. 40–90 AD) mentions the ophthalmological use of cuttlefish bone to treat leukoma in livestock, applying the powder obtained from it to the eyes of sick animals—Book II, chapter 21—[53,54]. In turn, in his encyclopaedic work *Naturalis Historia*, Pliny the Elder (23–79 AD) also indicated that this powder was able to cure cataracts (Book XXXII, chapter 71), although he did not specify whether it was a useful therapy for both humans and animals [55,56]. In the *Hortus Sanitatis*, the last encyclopaedic book on medical matters written in Latin and attributed to Johannes de Cuba [57], we can read that Aristotle stated that cuttlebone powders mixed with salt is a good remedy for curing “white ulcers” in the eyes of people and domestic animals [58].

In the same way [18], it is worth reconsidering our appreciation of the *Tresor de Beutat* (“Treasure of Beauty”), a medieval treatise devoted to female cosmetics and health. Attributed to Manuel Dias Calatayud, written in Catalan, and catalogued as Manuscript 68 of the Library of the University of Barcelona, it includes descriptions of over 200 formulas used as remedies, cosmetic advice and different types of treatment for women inhabitants on the Mediterranean coast of eastern Spain in the 14th century. A total of 223 products were identified, and of these, 47 (21%) were of animal origin belonging to 30 animals. Cuttlefish bone was used as powder to whiten the face or teeth [59].

This brief synopsis through history leads us to conclude that the therapeutic properties and uses of pulverised cuttlefish bone were well known in the ancient Greek world and early Byzantine times; in particular, it has been used in the treatment of various eye itches and diseases [5].

An interesting source for the diversity of remedies is the *Libro de las utilidades de los animales* (“Book of the Utilities of Animals”), possibly authored by Ibn al-Durayhim (1312–1361). This manuscript represents a stereotype of cultural eclecticism within the framework of traditions, integrating scientific, folklore and naturalistic knowledge [60]. Mother-of-pearl is said to be useful for heart palpitations, for fear produced by black bile and for lacrimation or tearfulness, keeping the nerves of the eye tight and invigorating them. The shell, burnt and mixed with honey, is good for hardness of the breasts. Preparations also include plasters for inflammations occurring at the base of the ears and with ashes, which are beneficial for red pustules appearing on the head. Other remedies are based on burning the shell in which the formation of the pearl has already been completed and spreading its ashes where there are gangrenous scrofulas, as it dries and cures them. Similarly, the burnt shell, if given as eye drops, cleanses the eye and rinses out the tears, doing well for the leucoma that occurs in it. If it is powdered and poured into the nose, it stops a nosebleed. Mixed with wet pine resin and smeared on alopecia, it cures it. If the shell is pulverised, mixed with duck fat and smeared on the breasts, it works well for the accumulation of milk. Pulverised with coriander and mixed with honey, it is effective for pustules and warts, and scabies in the womb. Cooked with oil and salt, seashells are good in cases of poisoning by venomous animals and for the treatment of chronic inflammations and infected ulcers, especially those on the nape.

It should be noted that, in addition to the shell, the animal itself is used. Thus, it is said that “if the animal from which the pearl originates is taken with its shells, boiled and the sauce, honey and wine are drunk and given to drink to those who have swelling without ulcers, it is good for them”. According to their beliefs, the fluids released by the animal are useful for gall bladder ulcers. Furthermore, if the animal is boiled in water with its shell and then taken out and burnt, and the face is smeared with its ashes, it purifies and beautifies it. The meat from the shell, pounded with honey and smeared where a rabid dog has bitten it, is beneficial [60].

It is also said that the sea snail, crushed and sieved, mixed with honey and lily, and drunk, is good for liver pain [60].

To conclude this period, we can comment on one of the most important writings of medieval pharmacy, which made it possible to prepare prescriptions with precision: the *Antidotarium Nicolai*. This manuscript was produced post-Constantinian with Arabic influence, and today, it still presents difficult and unresolved questions such as its dating and authorship. It is a very important antidotarium, which played a similar role to the herbarium *Ex herbis feminis* as a summary, selection and adaptation of the book *De Materia Medica* by Dioscorides, and with a clear pedagogical and didactic value [61,62,63]. Among the recipes we find pearls (perforated or not), which were used as astringents. Regarding pierced pearls, Fray Antonio Castell says that “Pearls are naturally not pierced except by the art and industry of lapidary artists, and it is noteworthy that pierced pearls are worse for medicine, because virtue is exhaled through the hole … The best are the thickest, clearest, roundest and softest pearls. The least good and virtuous are the smallest, such as are used in medicine …” [64]. In the *Antidotarium Nicolai*, the *Diamargariton* is inventoried [61] (p. 7), that is, an electuary based on pearls, which is also called *margarita* after the name in Greek [61] (pp. 2, 7).

### 3.2. The Modern Age: The Renaissance, the Baroque and the Enlightenment

As is well known, the Ottoman conquest of Constantinople in 1453, the invention of the modern printing press and the discovery of America are three great events that represent the beginning of modern science. Knowledge based on the explanation of natural phenomena through reason and experimentation is promoted.

In this sense, it is interesting to observe the influences and interactions between magical-religious medicine and scientific or empirical medicine. Thus, the Benedictine friar Martín Sarmiento, in his *Catálogo de voces y frases de la lengua gallega* (“Catalogue of voices and phrases of the Galician language”) (1746), states that the so-called *ollo de boy mariño* (“sea ox’s eye”), i.e., the striking orange-reddish calcified operculum of *Bolma rugosa* (Linnaeus, 1767) (Gastropoda, Turbinidae), is carried as an amulet for eyesight and erysipelas, and placed on the forehead stops any flow of blood [65] (p. 222). Undoubtedly, this text has great philological value, but also historical-ethnomedical interest, since through its compilations, the relationships between beliefs and science can be observed. In fact, the quotation of this amulet belongs to Anselmus de Boodt (ca. 1550–1632), also known as Anselmus Boëtius de Boodt, who was physician to Rudolph II of Habsburg [65].

In the Renaissance, it is important to emphasise the work of Saladino de Ascolo, *Compendium Aromatariorum* (1488), which includes the qualities that a professional pharmacist should have and his reference books, as well as the pharmaceutical works and how to obtain them. Although the work presents primarily scholastic features, its structure and some of its contents invite modernity and a break with medieval thought. In Spain, the step from medical botany to pharmacy would take place through the work of Pedro Benedicto Mateo [66]. Saladino de Ascolo includes pearls in his recipe book with cardiac virtues. He describes the electuary *Diamargariton*, noting that “it is a cordial confection of two kinds of pierced and unpierced pearls”. Castell would later contribute two different compositions bearing the name *diamargariton*: *Diamargantum simpl. incerti autoris seu electuario de margaritis* and *Puluis Diamargariti frigidi compo. incerti autoris*. Although they are preparations based on pearls, they differ in their composition, the first being much simpler; thus, the apothecary would prepare it with half an ounce of unpierced pearls and a pound of fine sugar melted in pink or bugloss or other cordial water. In the second case, *Puluis Diamargariti frigidi* states that the cardiac virtue of pearls is increased by ambergris, musk, gold leaf, ivory, precious stones, stag’s heart bone and raw silk. He makes another series of comments on how to improve the effectiveness of the preparation and discusses the formulas of other authors [64,67].

The physician and alchemist Paracelsus (1493–1541) advocated the use of the so-called *Anodyne balsam*, which had sedative properties and was prepared from pearls, opium, henbane, coral, amber, musk, deer antler, unicorn horn and bezoar stone [68]. In this context, the alchemist and professor of medicine at the University of Marburg in Hesse (Germany), Oswald Croll or Crollius (ca. 1563–1609) made interesting contributions to pharmacology in his work *Basilia Chymica*. In it, he recommends the use of *sal margaritarum* obtained by dissolving pearls in distilled vinegar and evaporating the solution to obtain calcium acetate [69]. In general, during the 16th century, the medical use of pearls was recommended for the treatment of haemorrhages, dysentery, irregular menstruation in women, various ophthalmic problems such as blurred vision, heart palpitations, or for whitening stained teeth [70].

Studies of modern therapeutics, whether during the Renaissance or the Baroque, show that pearls, seashells and cuttlefish bones have been used in all medical systems to fight or prevent disease [70,71,72,73]. It should be noted that cuttlefish bones and mother-of-pearl were used as an absorbent in Latin America, following the guidelines of the Jesuit apothecaries. The cuttlebone was used to make tooth powders, while the ground pearls were used to cure malaria, heal serious wounds and as a cardiotonic [71].

Spain in the 18th century needed an important cultural and scientific renovation in all areas of knowledge, and in medicine, the role of certain institutions outside the university world would be fundamental [74]. Thus, the importance of the Royal Colleges of Surgery (RCS) should be noted. The first of these was the RCS of the Navy in Cadiz (1748), followed by the RCS of Barcelona (1760), the RCS of New Spain (1768), the College of Surgery of San Carlos in Madrid (1780) and the College of Medicine and Surgery of San Fernando in Lima (1811) [75,76,77,78,79,80]. In this regard, it is essential to consider the medicines used by these surgeons. Thus, in the Pharmacopoeia of the Spanish Navy (Figure 2), mother-of-pearl was used as an absorbent powder. Among their virtues, they were said to precipitate acid humours, which abound in the stomach and intestines, and to calm stomach pains and flatus. However, it was not a simple medicine, but was made together with coral, crab eyes and chalk. The thick shells of *Magallana angulata* (Lamarck, 1819) (Bivalvia, Ostreidae), calcined, and mixed with calcined eggshells, cream of tartar, vitriolate of tartar and the substance called “saffron of Mars” (haematite), were used to treat constipation, evacuating the bowels, and as a diuretic. Cuttlebones were also recommended to treat scrofula, forming part of a complex formula along with figwort root (*Scrophularia* spp.), burnt sponge, pumice stone, white mushroom, white paper ash and halite or rock salt [81].

### 3.3. Marine Malacotherapy in the Contemporary Age

As we have commented above, one of the great references of medicine and pharmacy between the centuries and also in the 19th century were the Royal Colleges and the Academies of Medicine and Surgery, as well as the bibliographic collections of some Spanish faculties of Medicine. Bear in mind that in 1843, the Royal Colleges were eliminated and transformed into “Faculties of Medical Sciences”. Thus, for example, a large number of authors have analysed the archives and library of the RCS or those of the Faculty of Medicine of Cadiz (currently ceded to the Historical Library of the University of Cadiz) [82,83,84,85,86,87], which contain an exceptional and inexhaustible source of study on modern and contemporary Spanish medicine, surgery and therapeutics. In this respect, we can comment that Ignacio Ameller, who was Professor of the RCS of the Cadiz Navy and who in 1844 formed part of the first faculty of the Faculty of Medical Sciences of Cadiz, assessed the use that had been made of oyster shells up to that time. It was a lithontriptic medicine to destroy bladder stones [88], in which these seashells were combined with soap and calcined eggshells and land snail shells. However, he also noted that this preparation, which he describes as the “composition of Mademoiselle Hefens”, was falling into disuse in the experience of a majority of physicians who were by then beginning to consider it as ineffective [89,90]. Despite these views, it must be considered as a highly reputable medicine at the time, as evidenced by the purchase in 1753 of 157 small glass jars in which to place the shells, as recorded in the account books of the Real Colegio de Medicina y Cirugía de Cádiz (Royal College of Medicine and Surgery of Cadiz) [83,89,90]. There is no doubt that the substances we are studying were part of treatises and practical guides for doctors, surgeons or pharmacists until the 19th century.

In short, seashells were generally prepared by dissolving in vinegar and were a part of poultices, or they were ground to obtain medicinal powders, used as toothpaste and to treat dyspepsia, heartburn, leprosy or scrofula [91], but they were also used as a remedy against goitre and particularly against rabies due to their composition rich in calcium carbonate [92]. Pearls were considered a cordial remedy for being cold and hot at the same time, which is why they were used in cardiac tonics (*confortatio et laetificatio cordis*) [88]. However, the medications that had the greatest prestige were lithontriptics, stones dissolvers. In addition, from the formulas used by surgeons, during the 19th century, and as a legacy of the pharmacy of the Enlightenment, specific compositions were prepared, for example, the *agua de cal* (or lime water) prepared with oyster shells or with calcium carbonate extracted from mother-of-pearl shells. Basically, the idea was to provoke physiological reactions to regulate the acid–base balance with bicarbonates or carbonated waters, taking care to increase diuresis since the possibility of the formation of phosphate or carbonate stones when alkalising too much was known [88].

As for cuttlebone, it must be said that this therapeutic resource fell into disuse at a faster rate and, although it was also used as an absorbent and in the composition of certain eye drops, its use was practically relegated to cosmetics and the preparation of toothpastes [93]. Examples of these prescriptions can be obtained through the 19th century Spanish medical press. So, in *El Siglo Médico*, an important medical journal published in Madrid between 1854 and 1936, we can read, on 26 January 1890 (p. 60), about the usefulness and efficacy of tooth powders for dental hygiene in the first dentition, powders made from magnesium carbonate, medicated soap, pulverised cuttlebone and essence of mint.

## 4. Medicinal Properties, Modern Uses and Perspectives

Many mollusc-derived products have an extensive array of therapeutic properties, including antimicrobial, antioxidant, anticancer, anti-inflammatory, antihypertensive, wound healing and other medicinal properties [94,95,96,97]. As a result of a significant evolutionary divergence, the phylum Mollusca includes numerous and very diverse groups de marine invertebrates. Associated with this is their very significant chemical diversity. Marine molluscs, for example, use secondary metabolites to communicate and defend themselves against predators and microbial pathogens [98,99], and they have become the focus of many chemical studies aimed at isolating and identifying novel natural products. This highlights the need for further research to identify the chemical compounds of certain molluscan species to provide leads for novel drugs and biomaterials in the future.

### 4.1. Cuttlebone

Cuttlefish bone is composed primarily of aragonite (crystal forms of calcium carbonate) and *β*-chitin, and it has a very elaborate microstructure. Many sea invertebrates have very characteristic internal shells, but the cuttlebone is unique due to its chambered complex structure [100,101,102,103]. Commonly used as a calcium-rich dietary supplement for caged animals (birds, turtles) and as a grinding stone for the beaks of cage birds [104], cuttlebone combining high porosity with high compressive strength is a very interesting material from the biomimetic materials’ technology point of view [100,101,104].

#### 4.1.1. Pharmacological Activity and Pharmaceutical Applications

Cuttlebone is especially important in the field of promoting bone healing. Various studies reveal the potential of different biomaterials derived from cuttlebone to deliver 3D scaffolds with potential for bone formation and regeneration applications [105,106]. Cuttlebone-derived scaffolds are popular due to their biocompatibility and high regenerative potential.

Bone grafting is widely used to bridge major bone defects or to promote bone union. Concerning biocompatibility, fibrous capsules of hydroxyapatite from cuttlebone are a valuable bone graft material. Cuttlebone-derived hydroxyapatite, prepared from cuttlebone via hydrothermal transformation [107,108], is an appropriate biomaterial to stimulate bone formation and enhance bone regeneration [108]. For its effective osteoconduction, cuttlebone-derived hydroxyapatite is a safe material for use inside the body [109].

Alginate capsules with cuttlebone-derived fillers have been developed for bone repair applications. Prepared capsules are designed to be suitable for the treatment of small-sized bone loss provocative diseases, such as endodontic and periodontal diseases [110]. Proliferative and osteoconductive effects on the osteoblast-like MG-63 cells demonstrate the cellulose/cuttlebone scaffolds soaked in simulated body fluid as a favourable material for bone tissue engineering [111].

It has been discovered that cuttlefish bones are an excellent resource for producing desirable amounts of chitin and chitosan. Chitosan, a chitin-derived linear cationic polysaccharide, is a natural, hydrophilic, non-allergenic, biocompatible, non-toxic and biodegradable product obtained from chitin of cuttlebone [112,113]. Chemical modification of chitosan has been frequently carried out to prepare derivatives with applications in many fields including pharmaceutics [112,114,115,116]. For example, a low-molecular weight sulfated chitosan holds immense potential in carbohydrate-based pharmaceuticals [117], and it is a potential inhibitor of white spot syndrome virus proteins [118].

Likewise, antimicrobial activities of powdered cuttlebone have been demonstrated, and it can be used as an accessible natural source to provide novel, low-cost and safe antimicrobial agents [119]. For example, powdered cuttlebone has been found to be effective against the bacterium *Klebsiella oxytoca*, and antifungal activity against *Aspergillus flavus* has been also recorded [120].

Anti-inflammatory, immune-modulatory and wound healing activities of cuttlebone have been shown [97]. Cuttlebone is a valuable material for the treatment of skin wound such as ulcer lesions and burn injuries [121,122]. Cuttlebone extracts stimulated wound skins to induce acute inflammation and to promoted cell proliferation and matrix metalloproteases expression in fibroblast [122].

Prepared formulations from cuttlebone showed appropriate physicochemical properties and high antacid capacity. Then, as proposed by Mostoufi et al. [123], we can use formulations of cuttlefish bone as a good natural antacid drug that has fewer side effects, has high efficiency, is inexpensive, and is comparable with the marketed tablets and other antacid compounds.

#### 4.1.2. Environmental Utilization

The cuttlebone has a high environmental value. It can be useful in bioecological research, providing reliable information on environmental conditions. For example, the isotope composition of cuttlebone aragonite appears to be in isotopic equilibrium with the ambient seawater [124], and stable carbon and oxygen isotopes from it are natural tags for determining the degree of spatial connectivity between nearshore and offshore environments [125].

Cuttlebone is an effective bio-adsorbent, constituting an efficient, low-cost and eco-friendly technology for reducing copper pollution during wastewater treatment [126]. A superhydrophobic and oleophilic porous material obtained using biomass cuttlebone as a scaffold has an excellent oil–water separation efficiency [127]. This finding is important for the elimination of spills of crude oil and other marine organic pollutants, very common around the world and resulting in severe environmental and ecological damage.

In addition, cuttlebone powder is used in the production of green concrete. It led to the substitution of a portion of cement content, and consequently, this type of concrete causes less harm to environment [128].

### 4.2. Nacre

Nacre (mother-of-pearl), the inner lustrous and iridescent layer of many mollusc shells, is composed of more than 95% aragonite (a crystallographic form of calcium carbonate—CaCO_3_) and less than 5% of an organic matrix (consisting of proteins, glycoproteins, polysaccharides and lipids) [129,130,131,132], and it possesses a unique combination of remarkable mechanical strength, impact resistance and toughness. The excellent mechanical and biomedically desirable properties of nacre are related to its hierarchical structure and precisely designed organic–inorganic interface [130,132,133,134,135].

The microarchitecture of nacre has been classically illustrated as a lamellar “bricks-and-mortar” arrangement. The basic structural pattern is the assembly of oriented plate-like aragonite crystals with a “brick” (CaCO_3_ platelets of micron sizes and sub-micron thicknesses) and “mortar” (organic macromolecular component) organization [129,131,136,137,138]. The water present at the nanograin interfaces also contributes to the viscoelastic nature of nacre [129].

#### 4.2.1. Pharmacological Activity and Pharmaceutical Applications

Owing to this apparent simple morphology and peculiar properties, natural nacre has attracted considerable attention of chemists, biologists and material scientists and engineers. The current knowledge on microstructure and mechanics of nacre has favoured the fabrication of nacre-inspired artificial and related materials. The design and fabrication of de novo synthetic materials is an active area of research in mechanics of materials. A strong emphasis is given on the latest advances on the synthetic design and production of nacre-inspired materials and coatings, to be used in biomedical applications [130,135,137,139,140,141,142], and biomimetic strategies have been proposed to produce new layered nanocomposites in such a way that they produce the best result when interacting with the body [137,143,144,145]. For instance, artificial nacre-like coatings have been fashioned in a layer-by-layer approach using a lamination process. Several different methods have emerged to produce thin layer structures to enhance the mechanical behaviour of the individual components [146,147]. In addition, nacre is widely used as a system model to study biomineralization mechanisms [148,149], and in the future, nacre may have broad applications in biomineralization.

All of this demonstrates how nacre and nacre-derived materials show good interaction with bone, which makes this material attractive in the biomedical field, namely in the orthopaedic or dental areas. The field of bone tissue engineering requires materials capable of providing enhanced mechanical properties and promoting osteogenic cell lineage commitment. While bone repair has long relied almost exclusively on inorganic, calcium phosphate ceramics such as hydroxyapatite and their composites or on non-degradable metals, the organically derived shell nacre generated by molluscs has emerged as a promising alternative [131,132]. Studies, both in vitro and in vivo, have demonstrated nacre’s biocompatibility, biodegradability and osteogenic potential [142].

Nacre powder has a proliferation effect on osteoblasts, osteoclasts and bone marrow cells in the process of bone tissue formation and morphogenesis [142,150]. Nacre contains certain bone remodelling factors that activate osteoblasts and regulate protein signalling transduction to promote osteoblast mineralization [151], and one or more signal molecules capable of activating osteogenic bone marrow cells [152,153]. Likewise, nacre is well tolerated by the host tissue and stimulates a faster osteogenesis [154,155,156].

Nacre can be used directly as a bulk implant or as part of a composite material when combined with polymers or other ceramic products [131,132]. In vivo studies have shown that new bone is formed without causing any inflammation when nacre is implanted in the bone [157,158].

Bivalvia-derived nacre has recently gained interest as a potential alternative ceramic material in orthopaedic biomaterials, combining the integration and mechanical capabilities of calcium phosphates with increased bioactivity derived from proteins and biomolecules [131,132]. The potential of nacre as a versatile, bioactive ceramic capable of improving bone tissue regeneration and will elicit increased research efforts and innovation utilizing nacre.

Furthermore, in vitro studies have shown that water-soluble components extracted from the nacre promote the differentiation of preosteoblast cells and matrix mineralization [159]. Among the molecules that regulate the formation of nacre, p10 and p60 proteins promote these processes [160].

It was shown that nacre itself integrates well into bone tissue [161] and may stimulate the differentiation of stem cells into the osteoblast lineage [162,163]. The scientific basis of fusion with bone was first discovered in 1992 by Lopez et al. [164] and later confirmed by Lamghari et al. [165]. Under closer scrutiny, nacre was found to activate skeletal cells, induce bone formation and provide structural support in a human clinical trial [165]. The “water soluble matrix fraction” of nacre, despite the controversy concerning its definition according to nacre researchers, directly induces the formation of new bone [162]. Molecules from nacre matrix have been shown to decrease bone resorption by restricting osteoclast metabolism [151]. Furthermore, due to its organic content and plate-like design, nacre is mechanically tough, non-immunogenic and rapidly biodegradable, without eliciting detrimental physiological effects. These characteristics of nacre provides a unique substrate for delivery of a functional agent to sites of bone loss in quantities that lead to rapid bone repair and regeneration [132].

Nacre is a promising biomaterial in maxillofacial surgery [166]. Nacre has been proposed as a resorbable and osteoconductive material favouring bone apposition without triggering an inflammatory reaction [166], and it is biocompatible, non-toxic and biodegradable [131,156,166].

Nacre enhances tissue growth and bone tissue bonding. It also has excellent mechanical properties, such as resistance to fracture, that are like those of human bones [156,167]. Nacre powder combined with polylactic acid scaffolds could promote mouse bone marrow mesenchymal stem cell proliferation and increase alkaline phosphatase activity [142]. In the presence of a nacre extract, which can induce the early calcium precipitation in cells (after 7 days), the expression levels of osteogenic markers are higher than in normal with a dose-dependent manner [149]. Nacre extract is likely to stimulate MC3T3-E1 and osteoarthritic osteoblasts in vitro through containing ethanol-soluble diffusible factors. Thus, nacre extract can induce mineralization in osteoblasts [149,158]. However, no in vivo experimentations have been conducted owing possibly to the immunogenic properties of nacre.

In addition to the induction of bone formation, nacre powder promotes skin wound healing. In rat skin incisional injury models, nacre implanted in the dermis increased collagen synthesis by stimulating dermal fibroblasts [168]. Water-soluble nacre, with superior biocompatibility to it, enhanced wound healing recovery properties for burn-induced apoptotic and necrotic cellular damage and spurred angiogenesis [169]. When water-soluble nacre is applied to a burned area, the burn-induced granulation sites are rapidly filled with collagen, and the damaged dermis and epidermis are restored to the appearance of normal skin. Furthermore, water-soluble components from the nacre promote the proliferation of dermal fibroblast cells, enhance collagen secretion [168,169] and improve the healing process of burn wounds by rapidly restoring angiogenesis and fibroblast activity [142].

In several studies aimed at the development of dermatological applications, the in vitro and in vivo results showed that the expression of collagen, essential for healthy maintenance of skin, is increased when nacre is the basis of the treatment [170]. Agarwal et al. [171] found that powdered nacre shows limited cytotoxicity at high concentrations in scar-derived cells and exhibits no apparent oxidative stress on primary skin fibroblast cells and epidermal skin cells. Nacre powder extract induce the reconstruction of intercellular cements in cuticle and can be used to treat dermatitis symptoms [142]. The lipidic constituent of nacre stimulates a reconstitution of the intercellular content of the stratum corneum on atopic dermatitis [172] and moisturizes the skin [170]. For all this, nacre has become an ingredient of interest for skin-related cosmetics.

Correlated with the traditional use of nacre as a sedation and tranquilization agent, anticonvulsant and sedative-hypnotic activities of nacre powder are well known. Nacre original powder, nacre water-soluble protein, nacre acid-soluble protein and nacre conchiolin protein could down-regulate the expression of 5-HT3 receptor and up-regulate the level of GABAB [173].

The administration of nacre extracts improved scopolamine-induced impairment, namely, short-term memory, object recognition and spatial memory [174]. Treatment with nacre extract increases the expression of brain-derived neurotrophic factor and nerve growth factor, which decreased after scopolamine treatment [175]. 

Water-soluble matrix presents antioxidant activity. It has free radical scavenging ability and inhibits lipid peroxidation [142,159].

#### 4.2.2. Environmental Utilization

Since the excessive petroleum-based plastics entails a great threat to the environment and human health, it is highly desirable to develop new materials for plastic replacement. Inspired by a brick-and-mortar microstructure of nacre, nacre-mimetic sustainable structural materials are being manufactured [176].

## 5. Documentary Sources Selection Procedures

The immeasurability and dispersion of therapeutic resources throughout history generates a complex data universe. Hence to access the maximum number of documentary sources, a narrative qualitative systematic review of the most important international and national databases was conducted. Thus, in the first step, the query was made in the collection of bibliographic reference databases of Web of Science (1900–present). The survey included the three main areas of research: the Science Citation Index (SCI), the Social Sciences Citation Index (SSCI) and the Arts and Humanities Citation Index (A&HCI). The Cambridge Digital Library and the Anthropology Plus and JSTOR III—Arts and Sciences international databases were consulted. The national resources referenced include the database of Ph.D. Theses, TESEO; the information system of the databases of the CSIC (Spanish Research Council); the Dialnet bibliographic website; Google Scholar; and the catalogue of Public State Libraries. The overall search pattern covered the title, abstract and keywords concerning ethnozoology-related disciplines that have UNESCO codes (e.g., anthropology, the history of medicine, zoology) and the terms marine molluscs, seashells, shells, cuttlebone, cuttlefish bone, nacre, mother-of-pearl, folk medicine, folklore, ethnobiology, ethnozoology, ethnomedicine and zootherapy, in conjunction with the Spanish geographical context. No restrictions regarding the language of the publications consulted, although most relevant studies were published in Spanish. Finally, we have consulted the collections of the Historical Library of the University of Cadiz and its portal dedicated to the conservation of historical heritage, as well as the heritage collections of the Complutense Library of Madrid.

In addition, it should be noted that the humanistic work presented delves into biological taxonomy and animal species identification. Thus, after performing a thorough analysis of the references retrieved and studied, the data were included in a database with several fields to characterise the animal species used, the disease or condition treated, the geographical location of the remedy and its corresponding bibliographic citation. The vernacular names found were contrasted and subjected to discriminatory analysis following biological, ecological and biogeographical criteria. Regarding animal taxonomy and nomenclature, we followed the checklists included in the Catalogue of Life: COL (https://www.catalogueoflife.org/; accessed on 3 August 2023) and the World Register of Marine Species: WoRMS (https://www.marinespecies.org/; accessed on 29 July 2023).

## 6. Final Considerations and Reflections

As can be seen, the chosen material and method is in line with emerging paradigms where the interdisciplinary or multidisciplinary are giving way to the transdisciplinary through the humanities. In this sense, our work is positioned through discursive analysis from the ethnographic to the historical field to shed light on a biomedical issue related to the search for pharmaceutical resources.

Remedies based on marine animals have been little studied and almost all relegated to the world of curiosities. However, many of these resources, which may seem unusual in our European context, have survived through time, and can inspire ethnopharmacological works.

In addition, consider the importance of historical parallels and fluctuations between different medical systems. Hence our work also wishes to inspire new lines of work based on the identification of species at times when official medicine entered in the biological paradigm.

The work presented reveals the rich history of marine shells and cuttlebones through medicine and science. Moreover, the narrative used documents the evolution of popular and scientific knowledge linked to highly topical cultural and social aspects. We emphasise that these cultural patterns must be analysed and studied with a transdisciplinary approach, opening up new ways of understanding knowledge and science in a broad sense, that can solve global health and environmental problems, and contribute to the fulfilment of Sustainable Development Goals (SDG).

On the other hand, and from a pharmacological point of view, it can be confirmed that the relationship between the chemical composition of these animal derivative products and their medicinal properties is mainly due to the presence of calcium carbonate. For example, the use of this common substance as toothpaste allows the treatment of dental problems. By its physical action on the gums, which increases gingival irrigation, it promotes healing in the case of wounds and/or bleeding. There is no doubt that the cleansing and exfoliating action of calcium carbonate justifies many of its medical applications, which would not be without risk if not managed properly.

Among the documentation consulted, the influence of certain chemical principles in the preparation of remedies was observed. In this respect, we would like to highlight the use of ashes, since the combustion of calcium carbonate (CaCO_3_) gives rise to calcium oxide (CaO), which is transformed by hydration into calcium hydroxide (Ca(OH)_2_). This is why we can find justification for medical prescriptions, such as those used at the Royal College of Surgery of the Cadiz Navy, a cornerstone of the union of medical and surgical studies in Europe. As documented (see Section 3.2), those surgeons prepared and applied a very subtle powder made from seashells in complex prescriptions. The scientific basis of these recipes was discussed in the so-called *Observaciones de Juntas Literarias* (“Remarks of Literary Meetings”), the manuscripts of which should be reread and analysed. Therefore, at this time of the 275th anniversary of this institution, where different professional competences were brought together, we find it interesting to highlight the importance of philosophical eclecticism in science as a conceptual approach that harmonises theories and ideas derived from different fields in order to generate new ideas.

From our point of view, studies on medicine, pharmacy and society can shed light on the true meaning of natural pharmaceutical matter. A new look at polypharmacy without prejudices and a penchant for science could be useful to counteract negationist or fundamentalist positions. Hence, the history of medicine, anthropology and the humanities in general can be used as a source of inspiration and reflection for scientific progress in subjects such as those with which we are dealing.

It is very interesting that the great treatises of pharmacology, as well as the works prior to the biomedical paradigms, show formulas for the elaboration of medicinal lime hydrolats from seashells. Undoubtedly, the use of calcium salts in cosmetology is also in line with ancient treatises, both for their photoprotective character and for their use in face masks for skin whitening.

In general, we can point out that the pharmacological properties of the calcium carbonate of seashells and cuttlebones allowed a fairly wide effective medical use (e.g., as an antacid, anti-diarrhoea, alexipharmic, anti-gout, cardiac tonic, etc.), whose historical, cultural and anthropological view can open new epistemological paths in research with products of animal origin.

## Figures and Tables

**Figure 1 pharmaceuticals-16-01503-f001:**
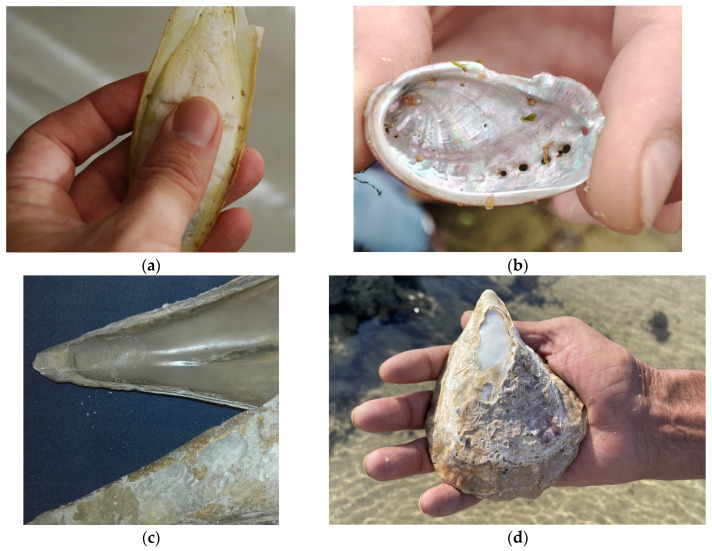
Examples of zootherapeutic resources of marine origin used in contemporary Spanish ethnomedicine: (**a**) internal shell of the cuttlefish; (**b**) inner iridescent “mother-of-pearl” layer of the ear-shaped shell of the green ormer (*Haliotis tuberculata*); (**c**) detail of the interior surface of the shell of the noble pen shell (*Pinna nobilis*) (photos by J.A. González); (**d**) worn-out shell of a large oyster in which the nacreous layer is visible (photo by J.R. Vallejo).

**Figure 2 pharmaceuticals-16-01503-f002:**
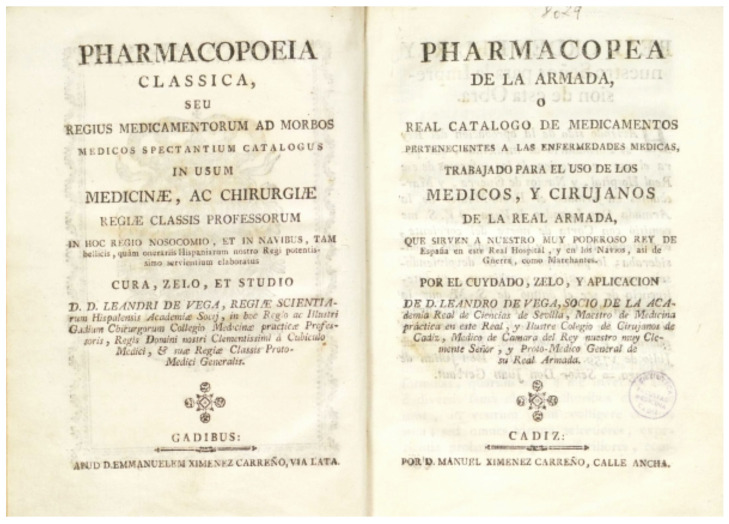
Cover of Pharmacopoeia of the Spanish Navy by D. Leandro de Vega [81]. Publication details: 165 pages; 21 cm (Historical Library of the University of Cadiz, Spain).

## Data Availability

Data sharing is not applicable.

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
