# Peer review of "The Use of Shells of Marine Molluscs in Spanish Ethnomedicine: A Historical Approach and Present and Future Perspectives"

_pharmaceuticals, 2023, doi:10.3390/ph16101503_

Round 1

Reviewer 1 Report

Comments to the Author

  1. The review paper provides a comprehensive overview of the role of marine molluscs shells in Spanish ethnomedicine throughout history and also covers modern transdisciplinary approaches for their usage. It will be a good addition to the field and will be very helpful for researchers in expanding their knowledge and understanding. The authors provide sufficient background information and include relevant references to support their discussion.
  2. Article has excellent title and good abstract.
  3. Thorough literature review covering various databases and disciplines and have also provided information about the systematic review process. Also, highlighted the importance of the review and research in this area by amalgamation of various knowledge resources.
  4. The review paper is generally well-organized, with clear subheadings that guide readers through different sections. However, there are some lengthy sections. For example, Page 10 (regarding applications of cuttlebone) and Page 11-13 (regarding nacre applications) is very detailed and lengthy. Can be separated into sub sections. Table classification or figure schematics can also enable an easier read.
  5. Line 94 and 298, provide English translation.
  6. Line 13, Line 28-30, Line 100, Line 100-102, Line 137-139, Line 176, Line 185, Line 208, Line 211-215, Line 243-247, Line 268-269, Line 309, Line 345, Line 394, Line 428, Line 540, Line 548-549, Line 616, check sentence for grammar and accuracy.
  7.  Few reference edits:

a) Lines 205 – 215, 602 – 604 and 616 – 621 overlaps with Ref 18 (J. Ethnobiol. Ethnomed. 2016, 12, 36) so statement should be rephrased.

b) Lines 508 – 510, 517 – 521, 550 – 552, overlaps with Ref 175 (Antioxidants 2021, 10, 505) so statement should be rephrased.

c) In the reference section, for Ref 1, Ref 2, Ref 3, Ref 8-17, Ref 19-30, Ref 34, Ref 36-42, Ref 45-50, Ref 53-56, Ref 58, Ref 60-86, Ref 88-93 English translation can be provided for easier read.

  1. Line 94 and 298, provide English translation.

In the reference section, for Ref 1, Ref 2, Ref 3, Ref 8-17, Ref 19-30, Ref 34, Ref 36-42, Ref 45-50, Ref 53-56, Ref 58, Ref 60-86, Ref 88-93 English translation can be provided for easier read.

Author Response

REVIEWER 1

Comments and Suggestions for Authors

  1. The review paper provides a comprehensive overview of the role of marine molluscs shells in Spanish ethnomedicine throughout history and also covers modern transdisciplinary approaches for their usage. It will be a good addition to the field and will be very helpful for researchers in expanding their knowledge and understanding. The authors provide sufficient background information and include relevant references to support their discussion. … We want to express our gratitude for your effort and time dedicated to reviewing this article… and thank you very much for your favourable comments. Your comments and suggestions have been of great help to us to improve our manuscript.
  2. Article has excellent title and good abstract. … Although another reviewer has suggested changing the title, reducing it more precisely, we have not made any such change. We think that this title clearly covers all facets included in the manuscript. Regarding the abstract, following the suggestion of another reviewer, we have reduced its length without losing information. Thank you for positively evaluating these points of the article.
  3. Thorough literature review covering various databases and disciplines and have also provided information about the systematic review process. Also, highlighted the importance of the review and research in this area by amalgamation of various knowledge resources. … The final section of the manuscript has been divided. Now there is a specific section related to the process of selecting documentary sources and the data collection.
  4. The review paper is generally well-organized, with clear subheadings that guide readers through different sections. However, there are some lengthy sections. For example, Page 10 (regarding applications of cuttlebone) and Page 11-13 (regarding nacre applications) is very detailed and lengthy. Can be separated into sub sections. Table classification or figure schematics can also enable an easier read. … OK, section 4 has been reorganised into subsections. Special attention has been given to pharmaceutical applications.
  5. Line 94 and 298, provide English translation. … OK, an English translation of the homeopathic principle has been included. We have not done the same with the formulas of line 298, since we believe that it is very appropriate, as in other cases, to respect the original name (which is difficult to translate).
  6. Line 13, Line 28-30, Line 100, Line 100-102, Line 137-139, Line 176, Line 185, Line 208, Line 211-215, Line 243-247, Line 268-269, Line 309, Line 345, Line 394, Line 428, Line 540, Line 548-549, Line 616, check sentence for grammar and accuracy. … OK, the text of these lines has been revised and rewritten.
  7.  Few reference edits:
  1. a) Lines 205 – 215, 602 – 604 (*) and 616 – 621 overlaps with Ref 18 (J. Ethnobiol. Ethnomed. 2016, 12, 36) so statement should be rephrased. … OK, the text in these lines has been rewritten. However, I would like to tell you that this is a paragraph (*) that we must include in each of our works reviewing the medicinal uses of animal groups, and it is not easy to write it in different ways.
  2. b) Lines 508 – 510, 517 – 521, 550 – 552, overlaps with Ref 175 (Antioxidants 2021, 10, 505) so statement should be rephrased. … OK, the text in these lines has been rewritten.
  3. c) In the reference section, for Ref 1, Ref 2, Ref 3, Ref 8-17, Ref 19-30, Ref 34, Ref 36-42, Ref 45-50, Ref 53-56, Ref 58, Ref 60-86, Ref 88-93 English translation can be provided for easier read. … This is a suggestion that has never been made to us. We have always written the original title of the works we have consulted. Potential readers will have no problem locating these references on the Internet. We do not see the inclusion of an English translation as necessary.

Reviewer 2 Report

I found the review carried out by the authors interesting, the only thing I suggest is to attach a section in which they describe the process of selecting the articles.

Minor editing of English language required

Author Response

REVIEWER 2

Comments and Suggestions for Authors

I found the review carried out by the authors interesting, the only thing I suggest is to attach a section in which they describe the process of selecting the articles. … OK, the final section (no. 5) of the manuscript has been divided. Now there is a specific section related to the process of selecting documentary sources and the data collection (5. “Documentary Sources Selection Procedures”). To maintain the narrative style of our review article, we have preferred to place this new section before the conclusions (following all the results and the discussion).

Comments on the Quality of English Language

Minor editing of English language required. … The entire text has been revised. At various points in the manuscript, based on the reviewers’ suggestions and comments changes have been made.

We want to express our gratitude for your effort and time dedicated to reviewing this article. Your comments and suggestions have been of great help to us to improve our manuscript.

Reviewer 3 Report

This manuscript explores the historical and contemporary medicinal uses of marine mollusc shells, particularly nacre (mother-of-pearl) and cuttlebone, in Spain. It delves into the rich history of these shells being employed in various forms of medicine, such as treating eye conditions, dental hygiene, skin issues, and even as amulets. The authors compare these traditional remedies with those found in classic works of medicine and provide insights into the chemical composition and medicinal properties of these products. Overall, the study is impressive but there are certain points that need to be addressed:

1.      The title is not appropriate. It should be revised and should be more precise.

2.      The elaboration of specific chemical compositions of nacre and cuttlebone is missing and how these compositions relate to their medicinal properties?

3.      Were there variations in the methods of preparing and using nacre and cuttlebone remedies across different regions of Spain, and if so, how did these variations impact their effectiveness?

4.      In the context of contemporary medicine, have there been any scientific studies or clinical trials that validate the historical uses of nacre and cuttlebone for various medical conditions?

5.      The potential risks or side effects associated with the use of nacre and cuttlebone in traditional remedies, especially when used for eye conditions or dental hygiene are missing.

Author Response

REVIEWER 3

Comments and Suggestions for Authors

This manuscript explores the historical and contemporary medicinal uses of marine mollusc shells, particularly nacre (mother-of-pearl) and cuttlebone, in Spain. It delves into the rich history of these shells being employed in various forms of medicine, such as treating eye conditions, dental hygiene, skin issues, and even as amulets. The authors compare these traditional remedies with those found in classic works of medicine and provide insights into the chemical composition and medicinal properties of these products. Overall, the study is impressive but there are certain points that need to be addressed:

  1. The title is not appropriate. It should be revised and should be more precise. … Another reviewer comments that the “article has excellent title”. We think that this title clearly covers all facets included in the manuscript. Our decision is to maintain the title as it was presented to the Guest Editors, and as it is incorporated in the submitted manuscript.
  2. The elaboration of specific chemical compositions of nacre and cuttlebone is missing and how these compositions relate to their medicinal properties? … Following your comments, we have elaborated a comment to deal with this issue in a new section 6.

The medicinal properties are justified by the presence of calcium carbonate, as is apparent from the information collected at the Royal College of Surgery in Cadiz and in folk transmission. For example, we commented that the use of carbonates can alleviate dental and gingival ailments since it can favour irrigation and therefore its healing and strengthening. It is also logical to think that calcium carbonate has an exfoliating power used since ancient times for the cleaning of the skin, eyelid roughness produced by keratosis or any type of pigmentary dermatosis. Other chemical compounds present in nacre or in cuttlebones would be negligible and would not have therapeutical character even due to a synergistic effect. It would be the case of elastic biopolymers that would depend on the different taxa and even the species.

At the time we consulted PubChem that as it knows is an open chemistry database at the National Institutes of Health (NIH) (see https://pubchem.ncbi.nlm.nih.gov/compound/10112). Despite your interest, we think it does not seem necessary to include information from this database.

  1. Were there variations in the methods of preparing and using nacre and cuttlebone remedies across different regions of Spain, and if so, how did these variations impact their effectiveness? … This is an interesting question about which there is no data. However, we have been studying popular medicine for years and we agree with authors like Anton Erkoreka. In 1993 this researcher already stated that there is only one European popular medicine, whereby extrapolation with his studies in the Basque Country (Spain). It is assumed that 50% of its practices have an origin in scientific medicine, 35% would be magical and religious remedies, 20% empirical and 5% come from primitive medicine. In this sense, we have commented on the discussions and remarks carried out in Cadiz by the surgeons in the so-called “Literary Meetings” or in the pharmacopoeias of the Navy (18th century) that are also transferred to the folk and there are no prescriptions that determine a greater or less effectiveness according to the mode of preparation.
  2. In the context of contemporary medicine, have there been any scientific studies or clinical trials that validate the historical uses of nacre and cuttlebone for various medical conditions? … We know the existence of commercial products based on natural calcium carbonate. However, there are no clinical trials of these natural products probably for reasons related to economic issues related to the pharmaceutical industry. If you consult a Vademecum of natural products such as Bronson Laboratories, Pharbest Pharmaceuticals or Sundown Naturals, you can find how the sale of Oyster Shell Calcium is quite frequent; however, there is no evidence based on clinical trials.
  3. The potential risks or side effects associated with the use of nacre and cuttlebone in traditional remedies, especially when used for eye conditions or dental hygiene are missing. … Yes, thank you very much. As we mentioned in point 2, we have added a comment on this but based on the benefit that could be obtained by increasing the irrigation of the gum. Evidently the mechanical action of poorly used calcium carbonate could have the negative abrasive effect of whether it is nacre or cuttlebone.

Finally, we want to express our gratitude for your effort and time dedicated to reviewing this article. Your comments and suggestions have been of great help to us to improve our manuscript.

Reviewer 4 Report

The authors summarized therapeutic effects of seashells on multiple disorders. The data on the chemical composition and medicinal properties of seashells used are provided; interestingly, they suggested that as sedation and a tranquilisation agent, nacre is an interesting source for further drug development. In general, the review is comprehensive and the manuscript was well-written. It may be published pending some minor revisions.

Detailed comments:

1) Mollusc species, disorders and the corresponding therapeutic effects should be listed in a table.

2) All known chemical compositions and medicinal properties of seashells should be listed in a table.

3) In Abstract, "eczema" usually develops on fingers, but not on face.

Some words are not scientific terms. For example, in Line 323, how to "comfort the heart"? And how to "heal mortal wounds"? What kinds of wounds are mortal?

Author Response

REVIEWER 4

Comments and Suggestions for Authors

The authors summarized therapeutic effects of seashells on multiple disorders. The data on the chemical composition and medicinal properties of seashells used are provided; interestingly, they suggested that as sedation and a tranquilisation agent, nacre is an interesting source for further drug development. In general, the review is comprehensive and the manuscript was well-written. It may be published pending some minor revisions. … We want to express our gratitude for your effort and time dedicated to reviewing this article… and thank you very much for your favourable comments. Your comments and suggestions have been of great help to us to improve our manuscript.

Detailed comments:

1) Mollusc species, disorders and the corresponding therapeutic effects should be listed in a table. //////////// 2) All known chemical compositions and medicinal properties of seashells should be listed in a table. … In other works reviewing the medical use of animals that we have published, we have included a table with the data you mention. Before submitting our manuscript, we thought about it quite a bit. We believe that in this case said table should not be included. Throughout the text, we mention a good number of species used, but it is not always easy to know which marine mollusc it is. The cuttlebone usually used is that of the Sepia offinalis species, but this is not the only cuttlefish caught on the Spanish coast. What we have done is include one more photograph, the one corresponding to the shell of a large oyster.

3) In Abstract, "eczema" usually develops on fingers, but not on face. … OK, thanks for the observation regarding this silly mistake. The abstract, following the suggestion of another reviewer, has been reduced in length (without losing information), and this word has disappeared.

Comments on the Quality of English Language

Some words are not scientific terms. For example, in Line 323, how to "comfort the heart"? And how to "heal mortal wounds"? What kinds of wounds are mortal? … The entire text has been revised. In this case, the two expressions that you highlight have been changed. We totally agree with your assessment, since these were two ancient concepts that, at the time, we limited ourselves to translating directly. Now the reader can read “to heal serious wounds and as a cardiotonic”.

Reviewer 5 Report

I have conducted a thorough review of the manuscript titled "The Use of Shells of Marine Molluscs in Spanish Ethnomedicine: A Historical Approach and Present and Future Perspectives." The authors have presented compelling and informative data regarding cuttlebone and nacre across diverse domains. The paper is deemed acceptable for publication; however, some corrections and modifications are necessary to enhance its overall quality and clarity.
1. This paper lacks a clear purpose or motivation.
2. This paper is deficient in a method section that delineates the procedures for acquiring the data utilized in the paper's exposition. However, the authors provided descriptions under the heading "5. Final Considerations and Reflections," which would be more appropriately positioned within the method section.

3. Under heading 4, "Cuttlebone and Nacre: Medicinal Properties, Modern Uses, and Perspectives," two principal themes are presented: cuttlebone and nacre. However, it is advisable for the authors to categorize the narrative information under each main theme into distinct subtopics. For instance, within the theme of cuttlebone, the information should be organized into three subtopics: pharmacological activity, pharmaceutical applications, and environmental utilization.
4. Within section 5, titled "Final Considerations and Reflections," the author should express informed opinions pertaining to the discussed topic, offering insights and suggestions for the future utilization of marine mollusk shells as indicated in the topic.

5. In the abstract section, it is imperative for the abstract to be succinct and effectively encapsulate all pertinent information.

In my view, minor editing of the English language is necessary.

Author Response

REVIEWER 5

Comments and Suggestions for Authors

I have conducted a thorough review of the manuscript titled "The Use of Shells of Marine Molluscs in Spanish Ethnomedicine: A Historical Approach and Present and Future Perspectives." The authors have presented compelling and informative data regarding cuttlebone and nacre across diverse domains. The paper is deemed acceptable for publication; however, some corrections and modifications are necessary to enhance its overall quality and clarity.

We want to express our gratitude for your effort and time dedicated to reviewing this article. Your comments and suggestions have been of great help to us to improve our manuscript.

  1. This paper lacks a clear purpose or motivation. … As mentioned in the final part of the manuscript, our review article is a historical, cultural and anthropological look at the medicinal use of certain products derived from marine molluscs. This work can open new epistemological paths in pharmacological research with products of animal origin.
  2. This paper is deficient in a method section that delineates the procedures for acquiring the data utilized in the paper's exposition. However, the authors provided descriptions under the heading "5. Final Considerations and Reflections," which would be more appropriately positioned within the method section. … OK, the final section (no. 5) of the manuscript has been divided. There is now a specific section related to the process of selecting documentary sources and the data collection (5. “Documentary Sources Selection Procedures”). To maintain the narrative style of our review article, we have preferred to place this new section before the conclusions (following all the results and the discussion).
  3. Under heading 4, "Cuttlebone and Nacre: Medicinal Properties, Modern Uses, and Perspectives," two principal themes are presented: cuttlebone and nacre. However, it is advisable for the authors to categorize the narrative information under each main theme into distinct subtopics. For instance, within the theme of cuttlebone, the information should be organized into three subtopics: pharmacological activity, pharmaceutical applications, and environmental utilization. … OK, the usage data collected in section 4 has been reorganised into subsections. In response to your pertinent comments, we have separated the data relating to the medicinal use of cuttlebone from those referring to the use of nacre. Likewise, for both cases, two subsections have been established (with your invaluable help): one that we have called “Pharmacological Activity and Pharmaceutical applications”, and another with the title “Environmental Utilization”.
  4. Within section 5, titled "Final Considerations and Reflections," the author should express informed opinions pertaining to the discussed topic, offering insights and suggestions for the future utilization of marine mollusk shells as indicated in the topic. … The new section 6 (“Final Considerations and Reflections”) has been expanded. A series of ideas and reflections have been included. We believe that it meets your comments and that potential readers will like it.
  5. In the abstract section, it is imperative for the abstract to be succinct and effectively encapsulate all pertinent information. … OK, the abstract has been reduced in length (we believe that without losing information).

Comments on the Quality of English Language

In my view, minor editing of the English language is necessary. … The entire text has been revised. Changes have been made at various points in the manuscript based on the suggestions and comments of all reviewers.

Round 2

Reviewer 5 Report

The authors kindly attended to my inquiries and suggestions aimed at improving the manuscript's structure. So, I strongly advocate accepting this manuscript for publication in Pharmaceuticals.